# Bis-allylic Deuterated DHA Alleviates Oxidative Stress in Retinal Epithelial Cells

**DOI:** 10.3390/antiox8100447

**Published:** 2019-10-01

**Authors:** Mélissa Rosell, Martin Giera, Philippe Brabet, Mikhail S. Shchepinov, Michel Guichardant, Thierry Durand, Joseph Vercauteren, Jean-Marie Galano, Céline Crauste

**Affiliations:** 1IBMM, Univ Montpellier, CNRS, ENSCM, 34093 Montpellier, France; 2Leiden University Medical Center, Center for Proteomics and Metabolomics, Albinusdreef 2, 2333ZA Leiden, The Netherlands; 3Institute for Neurosciences of Montpellier, INSERM U1051-UM, Hospital St Eloi, 80 rue Augustin Fliche, 34091 Montpellier, France; 4Retrotope, Inc., Los Altos, CA 94022, USA; 5Univ-Lyon, Inserm UMR 1060, Inra UMR 1397 (CarMeN Laboratory), IMBL, INSA-Lyon, 69100 Villeurbanne, France

**Keywords:** DHA, oxidative stress, kinetic isotope effect, lipid peroxidation, lipophenol, phenolipid

## Abstract

Oxidative stress plays a crucial role in developing and accelerating retinal diseases including age-related macular degeneration (AMD). Docosahexaenoic acid (DHA, C22:6, n-3), the main lipid constituent of retinal epithelial cell membranes, is highly prone to radical and enzymatic oxidation leading to deleterious or beneficial metabolites for retinal tissue. To inhibit radical oxidation while preserving enzymatic metabolism, deuterium was incorporated at specific positions of DHA, resulting in D_2_-DHA when incorporated at position 6 and D_4_-DHA when incorporated at the 6,9 *bis*-allylic positions. Both derivatives were able to decrease DHAs’ toxicity and free radical processes involved in lipid peroxidation, in ARPE-19 cells (Adult Retinal Pigment Epithelial cell line), under pro-oxidant conditions. Our positive results encouraged us to prepare lipophenolic-deuterated-DHA conjugates as possible drug candidates for AMD treatment. These novel derivatives proved efficient in limiting lipid peroxidation in ARPE-19 cells. Finally, we evaluated the underlying mechanisms and the enzymatic conversion of both deuterated DHA. While radical abstraction was affected at the deuterium incorporation sites, enzymatic conversion by the lipoxygenase 15s-LOX was not impacted. Our results suggest that site-specifically deuterated DHA could be used in the development of DHA conjugates for treatment of oxidative stress driven diseases, or as biological tools to study the roles, activities and mechanisms of DHA metabolites.

## 1. Introduction

Docosahexaenoic acid, (DHA, C22:6, n-3) is a polyunsaturated fatty acid that belongs to the omega-3 family. Several biological functions have been assigned to DHA, including anti-inflammatory [1], anti-angiogenesis [2] and anti-apoptotic roles [3]. One organ that heavily depends on DHA, is the eye. DHA is the most abundant polyunsaturated fatty acid (PUFA) of the retina cell membrane, representing more than 50% of the total fatty acid content within the rod outer segments of photoreceptors [4,5]. For this reason, this lipid is indispensable for retina development and for visual acuity. Overall, DHA plays a crucial role in preserving retina integrity and function. Mechanistically, DHA ensures fluidity of photoreceptor membranes, maintaining bilayer flexibility, thereby providing an adequate environment for conformational rhodopsin changes involved in its regeneration [6]. The mechanisms by which DHA protects retinal cells is still under investigation. To date, there is compelling evidence reporting the implication of enzymatically and non-enzymatically oxidized DHA derived metabolites (cell mediators and signaling molecules) in the activation of retinal pigment epithelium cell (RPE cells) repair mechanisms. Both pathways start with an abstraction of a hydrogen atom located at a *bis*-allylic position of DHA, either initiated by selective enzymes or by reactive oxygen species (ROS: ROO^•^, RO^•^, HO^•^ or ^•^NO_2_). Among the metabolites stemming from enzymatic oxidation, neuroprotectin NPD1 (a double oxygenation product of the enzyme 15-LOX, Scheme 1), is known to be an important pro-resolving mediator exerting anti-oxidant, anti-apoptotic and anti-inflammatory properties in RPE cells [7,8,9,10]. In parallel, DHA can also be oxidized by non-enzymatic mechanisms, serving as a scavenger of free radicals. It has been postulated that non-enzymatic oxidation products of DHA are involved in the modulation of gene expression causing a misbalance of the cells’ oxidative state. The level of oxidative stress will guide the nature and the concentration of metabolites formed and thus lead to a “healthy” or “toxic” environment resulting from the oxidation of DHA. Indeed DHA is prone to radical oxidation also leading to deleterious advanced lipid peroxidation end products (ALEs) such as for example 4-hydroxy-hex-2-enal (4-HHE) [11] or malondialdehyde [12]. Those reactive aldehydes, when formed in toxic concentration, affect the lipid membrane and cause protein/DNA damage in the cells [13,14]. In the particular case of retinal tissue (the DHA-richest tissue in the human body) exposed to high levels of light and oxygen, DHA peroxidation initiated by radical hydrogen abstraction at the 6 or, 6 and 9 *bis*-allylic positions, leads to 4-hydroxy-7-oxo-hept-5-enoic acid (HOHA, Scheme 1,2) [15]. This metabolite is prone to react with protein lysyl ε-amino residues, to generate 2-ω-carboxyethylpyrrole (CEP-adducts), found in high concentration in age related macular degeneration (AMD) patients [16,17,18]. CEP-adducts are endogenous factors known to induce angiogenesis in the retina (a pathological development that is associated with the advanced stages of AMD) [19] and to produce pro-AMD changes in animal models [20]. These results illustrate the real paradox between the “beneficial” and “harmful” effect of DHA, which probably depends on the degree of oxidative stress. 

As we gained interest in the development of an AMD or genetic macular degeneration (Stargardt disease) therapy, our team focused on the design of lipophilic antioxidants based on DHA conjugated to polyphenol (called lipophenol or phenolipid), able to inhibit carbonyl and oxidative stresses (COS) [21]. Different Stargardt forms are known. An autosomal dominant form (STGD3) is caused by mutations in the elongation of very long-chain fatty acids-like 4 (*ELOVL4*) gene affecting very long-chain polyunsaturated fatty acids (VLC-PUFA) metabolism [22]. However, the most common form of Stargardt’s disease (STGD1), is autosomal recessive, caused by genetic mutations in *ABCA4* gene, and leading to COS development and toxic lipid accumulation in RPE [23,24]. Indeed, regarding the aetiology of both genetic macular degeneration (STGD1) and AMD, COS mechanisms are implicated in the accumulation of a toxic *bis*-retinoid conjugate called A2E, in retinal pigment epithelium (RPE). Pathologic A2E formation occurs when *trans*-retinal (an electrophilic aldehyde molecule, that accumulates abnormally because of age or genetic mutation in *ABCA4* gene, in photoreceptor) is attacked by the nucleophilic primary amine of phosphatidylethanolamine (carbonyl stress) with subsequent oxidation (oxidative stress) [25]. Having several conjugated double bounds, A2E is especially susceptible to oxidative degradation leading to secondary toxic reactive aldehydes and epoxides [26,27]. *Trans*-retinal also exerts direct toxicity by increasing oxidative stress through overproduction of ROS [28,29].

Previously, we designed and validated the use of a lipophenolic derivative, an alkylated phloroglucinol conjugated to DHA. This agent activates antioxidant defense mechanism through the Nrf2/Keap1 pathway as well as free radical and carbonyl scavenging (anti-COS) properties (Scheme 2, IP-DHA). This lipophenol was designed to reverse the effects of defective *trans*-retinal clearance from the photoreceptor, in order to reduce A2E formation [21,30,31]. DHA was selected for several reasons; i.e., to increase phloroglucinol bioavailability in retinal tissue, since its high prevalence in retinal tissue indicates that this PUFA is efficiently transferred from blood to retina [32]; for its potential beneficial effect regarding its role as substrate for the formation of NPD1; and because clinical studies have proven that supplementation with high doses of Omega-3 lipids (EPA/DHA) was able to reduce AMD progression and positively influence functional parameters in patients affected by genetic Stargardt macular degeneration [33,34]. However, due to the Janus face of DHA, some concern remains using high doses of DHA or DHA-conjugates in the prevention of chronic diseases where oxidative stress plays a major role in toxic mechanisms [15].

In the present work, we hypothesized that specific chemical modulations of DHA will preserve its pro-resolving beneficial effect (protectin NPD1 formation, Scheme 1), while reducing its potentially toxic properties under oxidative conditions. A chemical modulation of DHA designed to reduce its radical oxidation vulnerability without affecting its enzymatic metabolism into beneficial protectins, was carried out in our previous synthetic efforts [35]. The developed deuterated DHA analogues were obtained by the introduction of deuterium selectively at the *bis*-allylic positions C-6 and C-9 (Scheme 2). As demonstrated by Shchepinov et al. [36,37,38,39,40] for other PUFAs, site-selective isotopic reinforcement at the *bis*-allylic sites prevents oxidative damage of PUFA due to a primary kinetic deuterium isotope effect, subsequently reducing radical induced lipid auto-oxidation by terminating/inhibiting the peroxidation chain process. Such deuterated PUFAs present normal incorporation and assimilation in animals, after oral supplementation [41]. Here, we aim to determine whether DHA selectively deuterated on positions 6 or 6 and 9 (both not involved in NPD1 biosynthesis), may protect RPE cells against oxidative stress. We report the effects of 6,6-D_2_-DHA and 6,6,9,9-D_4_-DHA (Scheme 2) in ARPE-19 cells (Adult Retinal Pigment Epithelial cell line) under pro-oxidant conditions. We highlight that the presence of deuterium is able to reduce radical induced oxidation stemming from *bis*-allylic hydrogen abstraction. Enzymatic metabolism to the protectin 10S,17S-diHDA (also called PDX) was not affected. In view of developing drug candidates for AMD treatment, lipophenolic-deuterated-DHA conjugates (Scheme 2) were also synthesized and evaluated showing beneficial effects related to deuterium incorporation. Our results suggest that it may be practical to use selectively deuterated-DHA derivatives in the development of DHA conjugates for macular degeneration treatment, or as biological tools to study role, activity and function of DHA metabolites.

## 2. Materials and Methods 

### 2.1. Chemicals

For cellular toxicity, C11-Bodipy assay and radical oxidation HPLC/MS analysis: Hydrogen peroxide solution (H_2_O_2_, 30 wt. % in H_2_O), all-trans-retinal, natural docosahexaenoic acid (DHA), 3-(4,5-dimethylthiazol-2-yl)-2,5-diphenyl tetrazolium bromide (MTT), 2,2′-azobis(2-amidinopropane) dihydrochloride (AAPH), Hexane (HPLC grade), formic acid and Dimethylsulfoxyde (DMSO, purity > 99.5%) were purchased from Sigma-Aldrich (St. Louis, MA, USA). N,N-dimethylformamide (DMF, purity > 99.5%) and Acetic acid (100%, liquid chromatography-mass spectrometry (LC-MS) grade) were purchased from Merck (Darmstadt, Germany). Methanol (LC-MS grade), acetonitrile (LC-MS grade), water (LC-MS grade) and isopropanol (LC-MS grade) were purchased from Fisher Scientific (Hampton, NH, USA). Ethyl acetate (purity > 99.8%) and EtOH (analytical reagent) were obtained from VWR (Radnor, PA, USA). Ammonium hydroxide (28–30 wt. % in water) was obtained from Acros organics (Morris, NJ, USA). The solid-phase extraction cartridges (Oasis Max, 3 mL, 60 mg) were purchased from Waters (Milford, MA, USA). 4,4-difluoro-5-(4-phenyl-1,3-butadienyl)-4-bora-3a,4a-diaza-s-indacene-3-undecanoic acid (C11-Bodipy^581/591^) was obtained from Molecular Probes (Eugene, OR, USA), dissolved in EtOH to prepare stock solution at 1 mM and stored at −20 °C in the dark. All DHAs and lipophenols were dissolved in dimethylsulfoxide (DMSO) to prepare a stock solution at 40 mM for cellular assays.

For enzymatic metabolism study: Tin(II)chloride, sodium tetraborate and 15-Lipoxidase from Glycine max (soybean) Type I-B as lyophilized powder (≥50,000 units/mg) were purchased from Sigma Aldrich (St. Louis, MA, USA). Methanol (LC-MS grade) was purchased from Merck. Water (LC-MS grade), acetic acid (LC-MS grade) and methyl formate (LC-MS grade) were purchased from Honeywell (Charlotte, NC, USA). The solid-phase extraction cartridges, Sep Pak^®^ Vac C18 (200 mg, 3CC) were purchased from Waters. DHA, DHA-d_5_, LTB4-d_4_, 15-HETE-d_8_ and synthetic PDX were purchased from Cayman Chemicals (Ann Arbor, MI, USA). D_10_-DHA was provided by Retrotope^®^ (Los Altos, CA, USA).

### 2.2. Chemical Synthesis

Deuterated DHAs, (6,6-D_2_)-DHA and (6,6,9,9-D_4_)-DHA (simplified respectively as D_2_-DHA and D_4_-DHA) were synthesized as previously described by Rosell et al. [35] Deuterated lipophenol, phloroglucinol-OiPr-O-(6,6-D_2_)-DHA and phloroglucinol-OiPr-O-(6,6,9,9-D_4_)-DHA (simplified respectively as IP-D_2_-DHA and IP-D_4_-DHA) were synthesized using the methodology described by Crauste et al. [21] Briefly, the coupling reactions between the protected silylated isopropylated-phloroglucinol and the deuterated DHAs were initiated using dicyclohexylcarbodiimide and dimethylaminopyridine (DCC/DMAP) as coupling reagents to access protected lipophenols. Final deprotection of triisopropylsilyl (TIPS) protecting groups by Et_3_N-3HF in dry tetrahydrofuran (THF) yielded deuterated lipophenols, IP-D_2_-DHA and IP-D_4_-DHA. A quality control assessment was established by a complete ^1^H and ^13^C NMR spectral analysis for each synthesized compound (chemical structure, general procedure, yield and NMR analysis are reported in Appendix A, Scheme 1). The internal standard (C21-15-F_2t_-IsoP) and NeuroP standard (4(RS)-F_4t_-NeuroP) used in MS/MS quantification of Neuroprostane (NeuroP), were previously synthesized in the laboratory according to published procedures [42,43].

### 2.3. Impact of Deuterium on DHA Toxicity and Free Radical Processes Involved in Cellular Lipid Peroxidation 

#### 2.3.1. Cell Culture

ARPE-19 cells were obtained from ATCC, and maintained in Dulbecco’s Modified Eagle’s Medium (DMEM)/Ham F12 (GIBCO) containing 10% (*v/v*) fetal bovine serum (FBS) and 1% (*v/v*) penicillin/streptomycin under a humidified (95%)/CO_2_ (5%) atmosphere at 37 °C. For splitting and experiments, cells were dissociated with 0.25% trypsin-EDTA, re-suspended in the culture medium and then plated at 1–3 × 10^5^ cells/mL. Cells were cultured and used up to a maximum of 10 passages. 

#### 2.3.2. Cell Viability

Cell viability was determined by the MTT colorimetric assay. Cells were incubated for 2 h with MTT reagent (0.5 mg/mL). The absorbance at 570 nm and 655 nm of individual wells was measured using a microplate reader (BioRad 550). The percentage of viable cells was calculated as [(OD570 sample – OD655 sample)/(OD570 control – OD655 control)] × 100%.

#### 2.3.3. Toxicity of Polyunsaturated Fatty Acid 

ARPE-19 cells were plated into 96-well plates (3 × 10^4^ cells/well) and cultured for 24 h to reach confluence before DHA treatment. The cells were treated in medium with 1% FBS (1% FBSM), or with H_2_O_2_ prepared in medium with FBS (1%, *v/v*) to a final concentration of 600 μM. Cells were then treated with DHA, D_2_-DHA and D_4_-DHA, at different concentrations (0–80 μm) for 24 h. Control cells were incubated with DMSO (0.2%). After 4 h, cell viability was determined in triplicate using the MTT colorimetric assay. Results were expressed in percentage of viable cells normalized with control conditions in the absence of PUFA and H_2_O_2_ stress. 

Sigmoidal doses-responses linked to the toxicity were obtained using GraphPad prism software, allowing to calculate IC_50_ values described as the concentration leading to 50% of cell viability. 

#### 2.3.4. Protection of Lipophenols against All-trans Retinal

ARPE-19 cells were plated into 96-well plates (3 × 10^4^ cells/well) and cultured for 24 h to reach confluence before lipophenol treatment. Cells were treated with serum free medium containing lipophenols at different concentrations (0–80 μM) for 1 h. Then all-*trans*-retinal was added to a final concentration of 25 μM for 4 h (in DMF), before rinsing with medium. Cell viability was determined 16–20 h later, in triplicate samples, using the MTT colorimetric assay. Control cells were incubated with DMSO (0.2%) and DMF (0.14%). The data are expressed as the percentage of untreated cells (CTL, without all-*trans*-retinal).

#### 2.3.5. Comparison of Lipid Peroxidation Status in ARPE-19 cells with C11-Bodipy^581/591^ under Oxidative Conditions

ARPE-19 cells were plated into 2 cm^2^ wells (2 × 10^5^ cells/well) and cultured for 24 h to reach confluence before PUFA or lipophenol treatment. The cell cultures were treated with 1% FBSM containing tested compounds (deuterated DHAs or deuterated lipophenols) at 50 μM for 24 h and then rinsed with media before the incubation of 5 µM C11-Bodipy^581/591^ for 30 min. Then, the cells were rinsed and treated with the different stressors, corresponding to different protocols. For chemical stress the cells were incubated with 400 µM of H_2_O_2_ during 24 h followed by FACS analysis. For photobleaching stress, the cells were exposed to a white LED lamp with an intensity of 5000 lux at room temperature during 1 h followed by the FACS analysis.

#### 2.3.6. Fluorescence-activated Cell Sorting Analysis

FACS was performed with the BD Accuri C6 Flow Cytometer and BD Accuri C6 Software for data acquisition (BD Biosciences). This flow cytometer is equipped with an excitation laser at 488 nm, and a fluorescence detector FL1 533/30 nm. Data were collected to 10,000 events for each sample using a flow rate of 35 µL/min. Parameters obtained for data analysis included: cell size from the forward scatter (FSC), granularity from the side scatter (SSC) and the green fluorescence of the stained cells with C11-Bodipy^581/591^.

First the SSC-A versus FSC-A dot-plot was used to gate the cellular population and remove the cellular debris. Then the SSC-A versus SSC-H dot-plot was used to remove cell doublets in the previous gate to give a final gate. The cellular population in the last gate designed was plotted on count versus log green fluorescence to give the fluorescence mean intensity of FL1. The mean FL1 values were normalized to the cells only stained with C11-Bodipy^581/591^.

#### 2.3.7. Statistical Analyses for Cellular Biological Tests

The data are presented as means ± SD determined from at least three independent experiments. In each experiment, all conditions were done at least in triplicate. Statistical analysis was performed by student’s *t*-test for gaussian distributions or by the non-parametric Mann Whitney test for non-normally distributed data (the normality of distributions was tested with a Shapiro-Wilk test) and differences with *p*-values < 0.05 were considered as statistically significant. When multiple comparisons were performed, one-way ANOVA analysis (Kruskal-Wallis test for non-normally distributed data) followed by Bonferroni (or Dunn) post-hoc test were used to evaluate statistical significance between groups. *p* < 0.05 was considered to indicate a statistically significant difference. 

### 2.4. Impact of Deuterium on DHA Oxidation in Non-cellular Media

#### 2.4.1. Oxidation Method of Natural/Deuterated DHAs

A solution of DHA in methanol (1 mg/mL, 0.5 mL) was added to 4.5 mL of phosphate-buffered saline solution (pH = 7.3) containing 1 mM of AAPH. The mixture was heated at 37 °C for 14 h, and then allowed to reach room temperature The mixture was spiked with 4 ng of internal standard (IS: C21-15-F_2t_-IsoP) and then purified using solid phase extraction.

#### 2.4.2. Solid Phase Extraction of Oxidized Samples 

For solid-phase extraction (SPE), Oasis MAX mixed polymer phase anion exchanger cartridges were used. Aliquots of 2 mL of sample were loaded on the cartridges previously conditioned with 2 mL of methanol and equilibrated with 2 mL of 0.02 M of formic acid (pH 4.5). After the sample was loaded, successive washing steps were performed using (i) 2 mL of aq. NH_4_OH 2% *(v/v)*, (ii) 2 mL of a mixture of methanol and aq. formic acid 0.02 M, pH 4.5 (3:7, *v/v*), (iii) 2 mL of hexane and (iv) 2 mL of a mixture of hexane and ethyl acetate (7:3, *v/v*). Compounds of interest retained on the column, were then eluted with two volumes of 1 mL of a (70:29.4:0.6; *v/v/v*) hexane/ethanol/acetic acid mixture. Finally, the sample was concentrated under a gentle stream of N_2_ at 40 °C. Recoveries were determined to range between 83 and 100% (see Appendix A).

#### 2.4.3. Liquid Chromatography/Mass Spectrometry for Neuroprostane Analysis

After SPE, samples were reconstituted with 100 μL of mobile phase ((A) water containing 0.1% (*v/v*) of formic acid, and (B) acetonitrile/methanol (80/20, *v/v*) with 0.1% (*v/v*) of formic acid, A/B (83:17, *v/v*)) and then injected.

An Eksigent (Sciex Applied Biosystems, Framingham, MA, USA) micro-LC equipped with a combi-PAL autosampler (CTC Analytics AG, Zwingen, Switzerland) was used. The autosampler vial tray was kept at 10 °C. Separation was performed by injecting 5 μL of sample onto a HALO C18 analytical column (100 × 0.5 mm, 2.7 μm; Eksigent Technologies, CA, USA). The flow rate was set at 0.03 mL/min. The column was held at 40 °C. Gradient elution was performed under the following conditions: from 0 to 1.6 min 17% solvent B; from 1.6 to 2.85 min %B increased up to 21%; from 2.85 to 7.27 min %B increased up to 25%; from 7.27 to 8.8 min %B increased up to 28.4%; from 8.8 to 9.62 min %B increased up to 33.1%; from 9.62 to 10.95 min %B increased up to 33.3%; from 10.95 to 15 min %B increased up to 40%; from 15 to 16.47 min %B increased up to 95%; between 16.47 and 18.92 conditions were held constant at 95% solvent B. Equilibration time was set at 2 min.

Mass spectrometry analysis was performed on an AB SCIEX QTRAP 5500 (Sciex). Electrospray ionization in the negative mode was applied (ESI-). The source voltage was set at −4.5 kV, and N_2_ was used as curtain gas. For analyte detection characteristic tandem MS fragments were used. For a detailed description of all setting, please see Appendix A. Quantification was accomplished using external calibration lines constructed with the internal standard (IS: C21-15-F_2t_-IsoP). The data are presented as means ± SD determined from 3 independent experiments, each performed in triplicate assay.

#### 2.4.4. Standard Solutions

Standards solutions of IS (C21-15-F_2t_-IsoP) and NeuroP standardt (4(*RS*)-F_4t_-NeuroP) used to determine the different characteristic transitions and the elution time for each compound were prepared in methanol at 1 µg/mL from 1 mg/mL stock solution in methanol. For calibration curves, standards solutions were prepared in methanol at the following concentrations, 1, 2, 4, 8, 16, 32, 64, 128, 256 and 512 ng/mL for 4(*RS*)-F_4t_-NeuroP. The volume of the internal standard (IS: C21-15-F_2t_-IsoP) used for the analysis of the samples (oxidized DHA) was 4 μL (from a stock solution at 1 µg/mL in methanol). Finally, calibration curves were calculated by the area ratio of the NeuroP and the internal standard.

### 2.5. Impact of Deuterium on DHA Enzymatic Oxidation by 15s-LOX

#### 2.5.1. Kinetics Parameters of 15s-LOX

The activity of 15-sLOX was monitored on a Beckman Coulter DU730 Life Sciences UV-visible spectrometer at a wavelength of 270 nm monitoring the conjugated triene structure in the dihydroxylation products. For the measurements a reaction mixture of DHA or deuterated DHA (0.02 mM; 0.04 mM; 0.1 mM and 0.2 mM) was prepared in 750 µL sodium borate buffer (50 mM, pH 9) by ultrasonification for 10 s. The reaction was subsequently initiated by the addition of 10 µL 15-sLOX (4 mg/mL) to make a total volume of 760 µL in a Hellma CEL2056 quartz cuvette with 1 cm path length (20 °C). The initial reaction rates (V_(0)_), derived from the recorded UV-absorbance using the law of Lambert-Beer (molar extinction coefficient = 40,000), were plotted against the respective DHA concentrations and fitted to Michaelis-Menten kinetics in GraphPad Prism 7.0. All V_(0)_ values were determined in triplicate and the average of three independent experiments was plotted against the substrate concentrations to yield Michaelis-Menten kinetics.

#### 2.5.2. Liquid Chromatography/Mass Spectrometry of 15s-LOX Oxidized Sample

In order to gain insights on the nature of the produced metabolites, LC-MS/MS analysis of the reaction products were carried out. To this end, the respective DHA variants (0.1 mM) were incubated in a 2 mL eppendorf tube with 10 µL 15-sLOX (4 mg/mL) in sodium borate buffer (50 mM, pH 9) for 30 min at room temperature and the reaction quenched by the addition of cold MeOH (760 µL). The hydroperoxides were reduced with the addition of aq. SnCl_2_ (50 µL, 5 mg/mL) and subsequently the mixture was centrifuged (16,100 ×g for 5 min). Water (2 mL) was added to the supernatant, followed by the dropwise addition of glacial acetic acid (40 µL). The products were purified by SPE using SepPak 200 mg C18 cartridges (3CC), according to published protocols [44]. The methyl formate eluates were evaporated to dryness, dissolved in MeOH (0.5 mL) and stored at −80 °C until analysis. The resulting solution was diluted 1000 times with MeOH for LC-MS/MS measurements. To 120 µL water, 72 µL of MeOH, 4 µL of the diluted solution and 4 µL of an internal standard (15-HETE-d8, Leukotriene LTB4-d4 and DHA-d5) was added. The samples were loaded in a SIL-30AC autosampler and the analytes separated on a Kinetex 1.7 µm C18 100Å, 50 × 2.1 mm LC column as described [45]. The detection of DHA metabolites was performed in selected ion monitoring mode using several traces in order to take the eventual loss of a deuterium atom into account. The following traces were used: *m/z* 327.2, 343.2, 359.2 and 375.2 for DHA and its metabolites, *m/z* 327.2, 328.2, 329.2, 344.2, 345.2, 360.2, 361.2, 376.2, 377.2 for D_2_-DHA, *m/z* 331.2, 345.3, 346.2, 361.2, 362.3, 363.2, 377.2, 378.2 and 379.2 for D_4_-DHA and *m/z* 337.2, 352.2, 353.2, 367.2, 368.2, 369.2, 383.2, 384.2 and 385.2 for D_10_-DHA. Tandem mass spectra were recorded as product ion scans. Quantification of the metabolites was done by calculating the area ratio between the analyte and the internal standard (DHA-d_5_ for non-hydroxylated, 15-HETE-d_8_ for mono-hydroxylated and LTB_4_-d_4_ for di-hydroxylated derivatives) using Multiquant version 3.0.2.

## 3. Results

### 3.1. Deuterium Incorporation at Bis-allylic Positions Decreases DHA Toxicity on ARPE-19 Cell Line

The impact of *bis*-allylic deuterium incorporation on DHA toxicity was studied using the ARPE-19 cell line. The toxicities of deuterated DHAs, corresponding to 6,6-D_2_-DHA and 6,6,9,9-D_4_-DHA, were compared to natural DHA under pro-oxidant conditions. This study was performed by an evaluation of cell survival after 24 h treatment with DHAs. Specific conditions of cell treatment have been selected to observe a toxic effect of DHA under stress, involving radical reactive species. The use of 1% FBSM (fetal bovine serum medium) was selected as primary stress condition for further experiments, providing elevated oxidative status and free radical processes involved in lipid peroxidation (Appendix A, DCFDA probe [46] and Appendix A C11-Bodipy^581/591^ probe) than using 2.5 or 10% FBSM. In parallel to stress induced by serum starvation, the study of DHAs toxicity was evaluated applying significantly stronger cellular stress conditions triggered by H_2_O_2_ treatment (600 µM) (Appendix A). At this concentration, we observed cell mortalities ranging from 60–70%.

The first observation was the high toxicity in ARPE-19 cells observed for respectively 60–65 µM, 65–70 µM and 70–75 µM of natural DHA, D_2_-DHA and D_4_-DHA, in serum starvation medium (Figure 1A). The cells presented a more elongated structure and less adherence between themselves. The dose-response curves associated to the toxicity of DHAs (Figure 1B,C) enabled us to calculate the half-maximal concentration leading to 50% of cell death (IC_50_, Table 1) under pro-oxidant conditions (1% FBSM or 600 µM of H_2_O_2_). 

When the cells were treated with DHA under serum starvation, the IC_50_ of deuterated DHAs, D_2_-DHA and D_4_-DHA, were significantly higher than natural DHA (65.21 ± 1.03 µM, Table 1, Figure 1B1) with respectively 69.48 ± 1.46 µM, and 75.60 ± 1.05 µM. Similar results were obtained for H_2_O_2_-stressed cells (Table 1, Figure 1C) where the IC_50_ values of D_2_-DHA and D_4_-DHA compared to natural DHA (65.47 ± 1.87 µM) were increased, to respectively 66.98 ± 1.89 µM and 76.24 ± 2.26 µM. However, only the IC_50_ value obtained using D_4_-DHA was significantly different from the value of natural DHA. 

Hence, deuterated D_4_-DHA was less toxic on ARPE-19 cells than natural DHA under pro-oxidant conditions, either under serum starvation or with toxic treatment of H_2_O_2_. The results obtained suggest that the incorporation of four deuteriums on *bis*-allylic positions significantly reduced the toxicity of DHAs under these stress conditions. 

Another interesting result stemming from the experiment using 1% FBSM, was the increased cell survival under DHA treatment. The dose-responses showed that DHAs were able to provide improvement in cell viability (Figure 1B2) between 0–60 µM, 0–65 µM and 0–70 µM respectively for natural DHA, D_2_-DHA and D_4_-DHA, corresponding to an increase of cell survival from 20 to 40%, compared to untreated cells (CTL). Deuterated-DHA allowed to increase this effect compared to natural DHA.

### 3.2. Deuterium Incorporation at Bis-allylic Positions of DHA Reduces Radical Lipid Peroxidation Status on ARPE-19 Cell Line

The impact of isotope reinforcement at *bis*-allylic positions of DHA was evaluated on radical induced lipid peroxidation using different pro-oxidant inducers. This study was performed using the fluorescent probe C11-BODIPY^581/591^, a sensitive indicator of free radical processes that have the potential to oxidize membrane lipids. Indeed, the lipid part of this probe allows its facile incorporation into the membranes [47]. Two forms are associated to the C11-BODIPY^581/591^ probe, a reduced and oxidized form that have two distinct fluorescence wavelength characteristics. C11-BODIPY^581/591^ is therefore not a quantitative methodology for lipid peroxidation, but gives information on free radical processes that possibly oxidize membrane lipids, and allows to compare lipid oxidative status between experiments [48]. This methodology was preferred over the thiobarbituric acid reactive substances (TBARS) method [49] in order to specifically focus on lipid oxidation. Moreover, fluorescence activated cell sorting (FACS) that showed a greater sensitivity compared to microplate fluorimetry [49], was used in this study.

ARPE-19 cells were subjected to three distinct stressors corresponding to serum starvation (1% FBSM) at two different durations (24 h and 48 h), treatment with 400 µM H_2_O_2_ (24 h), or photobleaching with white light at 5000 lux for 1 h (Figure 2, Figure 3 and Figure 4, Appendix A). Before the induced stress, cells were treated with 50 µM of DHA (natural or deuterated) for 24 h. Experiments were performed using non-toxic concentrations of DHA that had no impact on probe incorporation (Appendix A). Photobleaching conditions (duration and power) and H_2_O_2_ treatment (Appendix A) allowed preserving at least 70% of cell viability since FACS analysis focused only on living cells. 

When the cells were stressed by serum starvation (1% FBSM) for 48 h (Figure 2), preincubation with natural DHA caused an increase in lipid peroxidation status compared to untreated cells (CTL). A significant reduction of radical processes involved in lipid peroxidation was observed using incubation of both deuterated DHAs, especially for D_4_-DHA. An interesting result was obtained with D_4_-DHA treatment, which allowed to reach radical levels close to untreated cells (CTL). An increase of lipid peroxidation caused by H_2_O_2_ treatment (400 µM) was obtained compared to untreated cells (Figure 2, grey), showing that oxidation was also more pronounced under these conditions. An increase of oxidation was also obtained following incubation of natural DHA. As observed using serum starvation, treatment with both deuterated DHAs significantly impeded lipid peroxidation compared with natural DHA (Figure 2), and D_4_-DHA was more effective than D_2_-DHA. Finally, in assays focusing on lipid peroxidation, cell protection promoted by deuterated DHA was efficient under high stress conditions (H_2_O_2_) as well as under serum starvation.

Comparison of deuterated DHAs protection during different periods of serum starvation-stress (24 h and 48 h) was performed (Figure 3A,B). For both stress conditions, deuterated DHAs pretreatment allowed to decrease the level of lipid peroxidation compared to natural DHA independent of the stress duration. Moreover, D_4_-DHA further limited this oxidation when the cells were stressed during longer pre-treatment periods, 48 h (Figure 3B), whereas no differences between the two deuterated DHAs was observed at 24 h (Figure 3A).

Using photobleaching-stress (white light, 5000 lux, 1 h) a significant increase in radical processes involved in lipid peroxidation was obtained (Figure 4, grey). This shows high susceptibility of ARPE-19 cells to light induced oxidation. Furthermore, pretreatment with natural DHA lead to an increase in lipid peroxidation under photobleaching-stress. This lipid peroxidation induced by natural DHA treatment was significantly limited by both deuterated DHAs, with no difference between them. Both deuterated DHAs allowed to reach basal levels of radicals involved in lipid peroxidation obtained for control conditions. The same results were observed under similar experimental conditions (24 h of DHA incubation) replacing photobleaching-stress by serum starvation stress (Figure 4, white).

### 3.3. Deuterium Incorporation at Four Bis-allylic Positions of IP-DHA Lipophenol Reduces Lipid Peroxidation Status on ARPE-19 Cell Line

In view of therapeutic applications on AMD and Stargardt disease, we also evaluated the impact of deuterium on lipid peroxidation when incorporated in the lipid part of IP-DHA (Scheme 2), a lipophenol molecule already highlighted for its anti-carbonyl stress properties [21,30,31]. Indeed, lipid peroxidation is a consequence of oxidative stress occurring in the retina notably weakening RPE cells in AMD [50] by *bis*-retinoid A2E oxidation. We used the same probe and stress conditions previously described for DHAs: serum starvation (1% FBSM, 24 h and 48 h), H_2_O_2_ treatment (400 µM, 24 h) and photobleaching (white light, 5000 lux, 1 h) (Figure 5). 

First, applying serum starvation conditions for 48 h (Figure 5B), a slight but significant decrease in lipid peroxidation was observed for both deuterated IP-DHAs, IP-D_2_-DHA and IP-D_4_-DHA, compared to IP-DHA. Tetra-deuterated lipophenol was again more effective compared to its di-deuterated analogue. This protective effect is even more pronounced under serum starvation-stress for 24 h (Figure 5A), where IP-D_4_-DHA decreased lipid peroxidation compared to IP-DHA and IP-D_2_-DHA. Similar results were obtained for other stress conditions. When H_2_O_2_ was applied to the cells (Figure 5C), IP-D_4_-DHA decreased the lipid peroxidation status relative to IP-DHA and IP-D_2_-DHA. Finally, a reduction of lipid oxidation was also highlighted under photobleaching conditions (Figure 5D) when the cells had been pre-treated by IP-D_4_-DHA, against IP-D_2_-DHA and IP-DHA.

### 3.4. Deuterium Incorporation at Bis-allylic positions of IP-DHA Lipophenol Maintains Cytoprotection Against All-trans Retinal in ARPE-19 Cells

The cytoprotective ability of deuterated IP-DHA against all-*trans*-retinal was evaluated on ARPE-19 cells. Since the mechanism of IP-DHA protection is not fully elucidated, the purpose of this assay was to verify the presence of IP-DHAs protection against cytotoxic all-*trans*-retinal, with deuterated *bis*-allylic positions (Figure 6). Cells were first treated with IP-DHA in serum free medium for 1 h, followed by the incubation with a toxic concentration of carbonyl stressor, *trans*-retinal at 25 µM for 4 h. Finally, cell survival was determined after 20 h. Cell viability was increased (Figure 6) by pretreatment with IP-DHAs at 80 µM, respectively 42%, 52% and 47% by IP-DHA, IP-D_2_-DHA and IP-D_4_-DHA. Therefore, we achieved the same level of protection against *trans*-retinal cytotoxicity for all IP-DHAs, showing that the main cytoprotective property of the lipophenol was not altered by the presence of deuterium at *bis*-allylic positions.

### 3.5. Deuterium Incorporation at Bis-allylic Positions of DHA Impacts the Formation of Metabolites from Radical Lipid Peroxidation 

NeuroPs are specific metabolites of non-enzymatic lipid peroxidation, formed during the radical oxidation of DHA [51] and can be quantified in biological tissues. The different series of NeuroPs are defined according to their formation mechanism which involves an initial hydrogen atom abstraction by a free radical on a *bis*-allylic position of DHA. By studying specific F_4_-NeuroPs, the objective was to evaluate the influence of deuterium incorporation at *bis*-allylic position number 6 (D_2_-DHA), or 6 and 9 (D_4_-DHA). We followed the formation of the 4-F_4t_-NeuroP after DHAs radical oxidation, formed due to hydrogen atom abstraction at *bis*-allylic position number 6 (Figure 7). The formation of the 4-F_4t_-NeuroP was supposed to be negatively impacted by the presence of deuterium at *bis*-allylic positions 6. The DHA oxidation protocol was performed with the radical initiator AAPH in a mixture of phosphate buffer and methanol, at 37 °C during 14 h to form the desired NeuroP in agreement with Musiek et al. [52] After solid-phase extraction (SPE, Appendix A) and LC-MS/MS analysis of the mixture, a trend was observed corresponding to a decrease of 4-F_4t_-NeuroP formation from D_2_-DHA and D_4_-DHA, compared to natural DHA (Figure 7). Hence, deuterium incorporation seemed to limit the radical abstraction of hydrogen atoms at *bis*-allylic position number 6.

### 3.6. 15-sLOX Enzymatic Oxidation of Deuterated-DHA was not Inhibited by Deuterium Incorporation at the 6 and 9 Bis-allylic Positions

Enzymatic 15-sLOX metabolisation of DHA, D_2_-DHA and D_4_-DHA were compared to D_10_-DHA (DHA having all *bis*-allylic positions fully deuterated) (Appendix A). When comparing enzyme kinetics of 15s-LOX assessed for the double hydroxylation product, monitored at 270 nm, we found no substantial differences between DHA, D_2_-DHA and D_4_-DHA, while a significantly reduced activity was observed for D_10_-DHA. The V_max_ and K_M_ values of the investigated DHA derivatives are shown in Table 2. 

The main difference observed for the four compounds under investigation was a more than 10 fold decrease in the V_max_ value observed for D_10_-DHA. As the biosynthesis of protectin 10S,17S-diHDHA (PDX) involves two oxidation cycles by 15-sLOX, we gained interest in the product profiles of the different incubations. As can be seen from LC-MS analysis of enzymatically oxidized samples (Figure 8), the pre-dominant products detected for DHA, D_2_-DHA and D_4_-DHA was indeed the double oxygenation product 10S,17S-diHDHA with almost no substrate detectable after a 30 min reaction period. For D_10_-DHA however, the reaction towards the double oxygenation product was hampered, as can be evidentiated by a vast excess obtained for 17S-HDHA (first oxygenated product). This fact explains the observed slow reaction kinetics for the formation of the double oxygenation product and proves a strong deuterium effect obtained for the second oxygenation using D_10_-DHA. 

## 4. Discussion

DHA is the most abundant PUFA in the photoreceptor outer segment membrane [4,5], giving it a central role in visual function [50]. However, whereas this PUFA displays cytoprotective effects and appears to be indispensable for retinal function [3], a flip side of DHA is nowadays discussed, coming from its ability to be highly oxidized [53]. Indeed, toxification of DHA can occur through lipid peroxidation in cell membranes leading to toxic ALE [11], formation of reactive carbonyl species (RCS) and *in fine* denaturation of cellular substrates such as proteins and DNA [13,14]. DHA is highly sensitive to oxidation due to its five *bis*-allylic positions, prone to hydrogen abstraction and subsequently the formation of deleterious metabolites such as HHE (4-hydroxy-2-hexenal) [11] and HOHA, accountable for the accumulation of CEP adducts [16,17,18]. The latter were described as toxic derivatives responsible for angiogenesis induction and used increasingly as AMD biomarkers [19]. Hence, while DHA is indispensable for retina function, oxidative stress, favored by permanent exposition to light and high oxygen levels [50], might induce the formation of deleterious DHA metabolites. A proposed solution to reduce this toxic degradation of DHA is to reinforce its stability against oxidation using deuterium incorporation at *bis*-allylic positions [36,37,38,39,40]. We here evaluate the cellular benefits of selectively deuterated DHA either in free form or as lipophenol-DHA derivatives (IP-DHA, Scheme 2). The latter derivatives have already been shown to display cytoprotective activity on cellular and mouse models of retina disease [21,30,31]. Initially we assessed the toxicity of free deuterated DHA compared to natural DHA and studied their influence on membrane lipid peroxidation.

First we evaluated DHAs toxicity (Table 1, Figure 1) when ARPE-19 cells were subjected to either intermediary or acute pro-oxidant conditions (1% FBSM or toxic treatment with H_2_O_2_). As described by Halliwell [54] we observed that cell culture causes pro-oxidant conditions (Appendix A), particularly when using lower percentages of FBS. The acute stress induced by hydrogen peroxide is also described to promote senescence of ARPE-19 cells [55]. Toxic effects of DHA in ARPE-19 cells were previously shown by Liu et al. related to increased ROS levels and lipid peroxidation [56]. It confirmed that ARPE-19 cells can be weakened by DHA addition as also observed in our study. Treatment with DHA at increasing concentrations leads to higher cell mortality, but deuterium incorporation can partially rescue the cells. Since deuterium was not incorporated at the 18 *bis*-allylic position of DHA, harmful aldehydes such as HHE may still be produced using both deuterated DHAs. However, under both stress conditions, deuterated DHAs were less toxic than natural DHA corresponding to an increase of IC_50_ (concentration leading to the reduction of 50% of cell viability, Table 1, Figure 1B,C). In a *Saccharomyces cerevisiae* based model (coenzyme Q-deficient mutant yeasts which are sensitive to PUFA treatment), Hill et al. [36,37] showed a protection of deuterated PUFAs (α-linolenic and linoleic acids) against toxicity induced by lipid peroxidation either on mutants or wild type cells. In our case, we established sensitivity to DHA oxidation by reducing the amount of serum or increasing H_2_O_2_ concentrations. Our results also displayed the advantage to incorporate four deuteriums, since D_4_-DHA was less toxic compared to D_2_-DHA and natural DHA, probably because it was less prone to radical oxidation (Figure 1). Interestingly a dual role of DHA was observed during the toxicity assay using 1% FBSM as ROS inducer. Indeed, for the three employed DHAs, cell survival increased from 0 to 60 µM before toxicity became prevalent (Figure 1B2), confirming that DHA could be either beneficial for viability or toxic depending on the oxidation level (effect not observed using H_2_O_2_ stress). Moreover, the presence of deuterium allows to enhance cell viability (125% of survival using 70 µM of D_4_-DHA, Figure 1B2). A cytoprotective effect of DHA on ARPE-19 cells was reported by Johansson et al. [57], mediated through the activation of endogenous defense pathways such as the antioxidant response element (ARE) and the autophagy of damaged proteins. The medium in our experiments corresponding to 1% FBSM could probably promote the establishment of endogenous cell defenses coming from DHA metabolization favoring cell survival.

The impact of deuterium incorporation on free radical processes involved in lipid oxidation was evaluated using the fluorescent probe C11-BODIPY^581/591^ and FACS analysis [47,49]. Different stressors were applied to the cells during this study (Figure 2, Figure 3 and Figure 4) corresponding to serum starvation (1% FBSM), H_2_O_2_ treatment and white light exposure. Photobleaching stress exposure was chosen because stress coming from light exposure is involved in the mechanism of toxification of *bis*-retinoid (A2E) through photo-oxidation in the physiopathology of retina diseases [25,58]. As described by Liu et al. [56] we showed an increase of radical involved in lipid peroxidation in ARPE-19 cells after DHA treatment followed by white light exposure (Figure 4). This was also observed using serum starvation and H_2_O_2_ treatment (Figure 2). Both deuterated DHAs showed protection against lipid peroxidation generated by incubation of natural DHA under all stress conditions investigated. The incorporation of four deuteriums in DHA allowed to reach levels of radical involved in lipid peroxidation of untreated cells under serum starvation in both protocols (Figure 2 and Figure 4, white).

Furthermore, we observed a difference between deuterated-DHA protection when pro-oxidant conditions were applied during 24 or 48 h. Using a period of 24 h of stress, both D_2_-DHA and D_4_-DHA decreased lipid peroxidation status with the same efficiency, reaching oxidation levels of untreated cells (Figure 4, vs. CTL). Similarly, Hill et al. [37] reported, using deuterated α-linolenic acid (ALA), that additional deuterium incorporation does not necessarily correspond to a limitation of lipid peroxidation. Indeed, deuterium incorporation in at least one *bis*-allylic position seems to be sufficient to impede this deleterious mechanism. However, our results suggest that D_4_-DHA displayed a better protection compared to D_2_-DHA when the cells incurred serum starvation-stress during longer periods (24 h versus 48 h, respectively Figure 3A,B). It seems that the longer the cells were subjected to oxidation, the higher is the protection with increasing deuterium incorporation.

The link between toxicity and lipid peroxidation remained intricate in this study. We observed that under serum starvation, DHA, D_2_-DHA and D_4_-DHA did not induced cell mortality at 50 µM (Figure 1B2). At this same concentration, treatment of ARPE-19 cells with natural DHA considerably increased the lipid peroxidation process. A first observation could be made: depending on the strength of the oxidation process, lipid peroxidation is not necessarily correlated to DHA toxicity. In addition, deuterated D_4_-DHA, although able to increase cell viability compared to untreated cells at 50 µM (Figure 1B2), it was also able to reduce radicals involved in lipid peroxidation to reach a basal level compared to control cells (Figure 2). Finally, the beneficial effect observed on cell survival at 50 µM DHAs is not due to oxidized metabolites coming from a strong free radical oxidation process. In addition, both deuterated DHAs were less toxic than natural DHA, and both were able to reduce lipid peroxidation. Hence, this decreased toxicity compared to natural DHA could be linked to the limitation of intensive lipid peroxidation due to deuterium incorporation at *bis*-allylic positions. Indeed, this isotope reinforcement could preserve the cells against deleterious oxidative mechanisms probably in a direct way by decreasing the sensitivity of DHA to radical oxidation or indirectly by impeding lipid peroxidation of natural PUFAs present in the cells. However, we can hypothesize that toxicity induced by lipid peroxidation is a complex mechanism probably involving various PUFA metabolites (peroxides, aldehydes) whose proportion and toxicity depend on DHA oxidability and degree of peroxidation.

The impact of deuterium incorporation to reduce lipid peroxidation status was also evaluated for lipophenol molecules. IP-DHA was previously highlighted to limit oxidative and carbonyl stress (deleterious mechanisms involved in AMD and Stargardt disease [21,30,31]). Hence, it was interesting to compare the capacity of IP-DHA, IP-D_2_-DHA and IP-D_4_-DHA (Scheme 2) to limit specific lipid peroxidation, one of the consequences of the pathological mechanisms involved in AMD [16]. As previously observed for free DHAs but in relatively lower proportion, deuterated IP-DHAs displayed a reduction of free radical involved in lipid peroxidation compared to IP-DHA treatment (Figure 5). The difference of protection observed between free deuterated PUFA and deuterated IP-DHA lipophenol could be due to a different rate of membrane incorporation. Only IP-D_4_-DHA showed a significant protection against this mechanism under the three stress conditions. In addition to its possible ester cleavage liberating the more protective D_4_-DHA, the IP-D_4_-DHA lipophenol appears as a promising compound able to reduce lipid peroxidation when the cells are subjected to pro-oxidant conditions, without losing its protection against carbonyl stress (*trans*-retinal seen in Figure 6).

To investigate the connection between cell protection provided by deuterated DHA and reduction of radical oxidation, the formation of F_4_-NeuroPs (Figure 7) formed during non-enzymatic radical oxidation of natural and deuterated DHA, was studied. The NeuroP of the series 4 is described to be among the predominant NeuroPs observed in in vivo studies [59]. The isotope reinforcement of DHA at position 6 (D_2_-DHA) or 6 and 9 (D_4_-DHA) was supposed to limit the formation of NeuroPs through the prevention of radical induced lipid peroxidation of DHA, especially of 4-F_4t_-NeuroP which is initiated by hydrogen abstraction at the 6 position. This was confirmed, since the formation of 4-F_4t_-NeuroPs was lower after oxidation of D_2_-DHA and D_4_-DHA in comparison to DHA oxidation (Figure 7). Deuterium incorporation impedes radical oxidation of DHA at the *bis*-allylic position 6, through the kinetic isotope effect described by Shchepinov et al. [60]. Cleavage of the D-C bond is slowed down compared to the H-C bond. This result suggested a protective effect of deuterium incorporation against hydrogen abstraction, the first step of DHA lipid peroxidation leading to deleterious metabolites such as HOHA (in case of retina pathologies). One can assume that this process would be present in a cellular surrounding, where the presence of the D-C bond would break the chain reaction of a radical lipid peroxidation process taking place repeatedly all along the bilayer membrane. Thus, blocking the global oxidation propagation might be more important when the lipids are arranged into a bi-lipid layer, due to spatial proximity of the PUFA chains, as was shown in the work of Hill et al. [37] during the evaluation of protective effects of deuterated linoleic and α-linolenic acid in living systems. We observed similar effects in a cellular model; global lipid peroxidation status was reduced using deuterated DHA that might be partially incorporated in the ARPE-19 cell membrane (Figure 2 and Figure 4).

A final verification of the advantage of deuterated DHAs was investigated to clarify their susceptibility to enzymatic oxidation. We aimed to investigate the enzymatic oxidation of DHA by the stereospecific abstraction of a hydrogen atom at the 15 position. This position is the first oxidation location during the enzymatic transformation of DHA into protectins such as NPD1 or PDX. NPD1 is a potent RPE cell mediator involved in cell-protection, anti-inflammatory responses, prosurvival repair signal and induction of anti-apoptotic enzymes, responsible for some of the advantageous characteristics of DHA [7,9]. Enzymatic oxidation leads to deuterated protectin PDX from both D_2_-DHA and D_4_-DHA. As expected, deuterium at positions 6 or 9 only marginally affected the affinity of DHA to the 15s-LOX enzyme (soybean lipoxygenase 1-type B) compared to deuterium incorporation in all *bis*-allylic positions (D_10_-DHA) (Table 2). Presence of 17S-HDHA (Figure 8) during enzymatic oxidation related to the possibility of deuterated DHA being enzymatically oxidized at the 15 position. In contrast to selected deuterated DHAs, D_10_-DHA, having deuterium at the 15 *bis*-allylic position, has a limited biotransformation to PDX using 15s-LOX, due to an isotope kinetic effect, as already observed in a deuterated LA analogue [61].

This underlines the importance of the “selective” deuteration of PUFA when developing biological tools or new deuterated therapeutic tools, which allows access to selected oxidized metabolites, in particular, in the case of retina pathologies. However it is important to note that used as PUFA supplementation, the protective actions of deuterated PUFA against lipid peroxidation have been reported even at only 20% of incorporation [37], leaving a large proportion of natural PUFAs in the membrane, free to be enzymatically oxidized.

## 5. Conclusions

In summary, our work demonstrates the efficiency of selective deuterium incorporation at *bis*-allylic positions of DHA, to decrease toxicity and lipid peroxidation in a retina cell line, under oxidative conditions. The association of deuterated DHA with alkyl-phloroglucinol allows such lipophenol to acquire efficiency against lipid peroxidation in addition to its anti-carbonyl stress activity. Moreover, deuterated lipophenol may probably be cleaved by esterases to liberate free deuterated DHA able to efficiently protect and preserve the tissue against lipid peroxidation. In vitro cell free studies of radical oxidation suggested that deuterium incorporation decreased radical abstraction at the site of deuteration. As a next step, CEP production should be assessed in combination with deuterated DHA treatment. Moreover, enzymatic oxidation by 15s-lipoxygenase giving rise to protectins was maintained using selectively deuterated DHA. In addition, since in the cell, the position of hydrogen abstraction varies depending on the identity of the oxygenase enzyme, it would be interesting to evaluate the impact of the presence of specific *bis*-allylic deuterium (on DHA) using a mixture of various enzymes (i.e., LOX and COX enzymes) and to identify the variability of the metabolite profile correlating with selective positions of deuterium. As already shown by Dennis et al. on deuterated arachidonic acid [62], this process would lead to specific DHAs able to be preferentially oxidized into resolvins, such as neuroprotectins or maresins, or into specific series of NeuroPs. Such a deuterated DHA library would be an efficient tool to evaluate the potential activities of the various DHA metabolites.

Efficient in reducing oxidative stress arising from radical sources, deuterated DHA would be an interesting tool to study the role, the origin and the target of oxidative stress in retina pathologies, as performed by Donato and coworkers in retinitis pigmentosa [63], using transcriptomic studies. Gene expression changes could be studied under oxidative stress in ARPE-19 with or without deuterated DHA treatment. Since oxidative stress may also come from enzymatic dysfunction [64], the relation between the different kinds of oxidative stress (enzymatic, radical or both) is an important point to study in order to better understand the etiopathogenesis of retina pathology.

At present, no approved pharmacologic drug for the treatment of dry AMD is available. The current study provides novel derivatives able to reduce both carbonyl stress and lipid peroxidation in ARPE-19 cells, both processes involved in progression of the disease. Deuterated DHA represents an interesting biological tool to study DHA biology and to develop new promising drugs for dry AMD. In addition, since all neuronal tissues contain high levels of DHA, it will be important to assess the therapeutic potential of deuterated lipophenols, for other neurodegenerative diseases (Alzheimer’s, Parkinson’s) also involving carbonyl and oxidative stress mechanisms.

## 6. Patents

Deuterated IP-DHA lipophenols presented in this work are described in the patent: P. Brabet, D. Cia, L. Guillou, C. Hamel, C. Vigor, T. Durand, C. Crauste, J. Vercauteren, New lipophenol compounds and uses thereof, (2015) WO2015162265A1.

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
