# Peer review of "Bis-allylic Deuterated DHA Alleviates Oxidative Stress in Retinal Epithelial Cells"

_antioxidants, 2019, doi:10.3390/antiox8100447_

Round 1
Reviewer 1 Report
Rosell et al. realized a very useful overview depicting the possible use of site-specifically deuterated DHA in the development of DHA conjugates for treatment of oxidative stress driven diseases, like AMD. I consider the manuscript very exhaustive and fascinating but, in the same time, I suggest several revisions to complete and further improve the global quality of the paper:
· When the authors speak about Stargardt disease and COS involvement in its etiology, they add only references about ABCA4gene. A more recent paper by Donato et al. regards the identification of early mechanisms of neural cell survival mediated by DHA-synthesized ELVs, as one produced by ELOVL4 protein [PMID: 29417145]. I suggest to add this reference to the proper paragraph in Introduction section.
· I suggest the authors to add a “Future perspectives” subsection, explaining how their experiment could reveal a real translational approach. One of them could be the evaluation of the relationship of several oxidative stress – related enzymes and deuterated DHA treatment. I suggest the authors to add references about one of them, the glyoxalase I (GLO1), involved both in oxidative stress and apoptotic pathway in eye-related diseases, like retinitis pigmentosa [PMID: 30099685].
· Another interesting point that I consider very useful to discuss in your manuscript regards a possible transcriptomic study able to highlight the possible gene expression changes of RPE cells treated by deuterated DHA. I suggest the authors to insert several references about a robust workflow followed for such kind experiments, as already realized for retinal pigment epithelium treated by ox-LDL [PMID: 29435412; PMID: 30413775; PMID: 30569795].
· Page 6: in “Statistical analyses for cellular biology tests” section must be considered the use of a post-hoc test (e.g. Bonferroni or Holm-Bonferroni corrections) to give strength to the resulting data. Why no one was applied?
· Even if the manuscript is well written, it requires minor English revisions.
Author Response
Reviewer 1:
First, we thank reviewer 1 for his/her time in reviewing this manuscript. The comments were much valuable and we have revised the manuscript accordingly and as much as possible. For more clarity, changes related to all reviewers’ comments were highlighted in green in the manuscript; changes related to English revisions were highlighted using “Track Changes" function in Microsoft Word.
Rosell et al. realized a very useful overview depicting the possible use of site-specifically deuterated DHA in the development of DHA conjugates for treatment of oxidative stress driven diseases, like AMD. I consider the manuscript very exhaustive and fascinating but, in the same time, I suggest several revisions to complete and further improve the global quality of the paper:
When the authors speak about Stargardt disease and COS involvement in its etiology, they add only references about ABCA4gene. A more recent paper by Donato et al. regards the identification of early mechanisms of neural cell survival mediated by DHA-synthesized ELVs, as one produced by ELOVL4 protein [PMID: 29417145]. I suggest to add this reference to the proper paragraph in Introduction section.We agree with reviewer 1, we added this reference and a sentence in the introduction to highlight the difference between recessive STGD1 and dominant STGD3. In our work we are more interested in STGD1 since STGD3 do not involved COS mechanism as principal toxic mechanisms, and also because, according to literature, ELOVL4 is not directly involved in DHA metabolism (RE Anderson and colleague’s, 2009 Adv Exp Med Biol, 2010 Mol vis, 2012 J Lip Res).
I suggest the authors to add a “Future perspectives” subsection, explaining how their experiment could reveal a real translational approach. One of them could be the evaluation of the relationship of several oxidative stress – related enzymes and deuterated DHA treatment. I suggest the authors to add references about one of them, the glyoxalase I (GLO1), involved both in oxidative stress and apoptotic pathway in eye-related diseases, like retinitis pigmentosa [PMID: 30099685]. Another interesting point that I consider very useful to discuss in your manuscript regards a possible transcriptomic study able to highlight the possible gene expression changes of RPE cells treated by deuterated DHA. I suggest the authors to insert several references about a robust workflow followed for such kind experiments, as already realized for retinal pigment epithelium treated by ox-LDL [PMID: 29435412; PMID: 30413775; PMID: 30569795].Regarding both last points, a sentence was added in the conclusion section to highlight the possible perspectives that could be envisaged using deuterated-DHA, for studying the role, the target, the different kind of oxidative stresses in retina pathologies. References of Donato and coworkers were added in this paragraph:
“Efficient in reducing oxidative stress arising from radical sources, deuterated DHA would be an interesting tool to study the role, the origin and the target of oxidative stress in retina pathologies, as performed by Donato and coworkers in retinitis pigmentosa [Donato et al. Cell Cycle 2019], using transcriptomic studies. Gene expression changes could be studied under oxidative stress in ARPE-19 with or without deuterated DHA treatment. Since oxidative stress may also come from enzymatic dysfunction [Donato et al. Mol. Biol. Rep. 2018], the relation between the different kinds of oxidative stress (enzymatic, radical or both) is an important point to study in order to better understand the etiopathogenesis of retina pathology”.
Page 6: in “Statistical analyses for cellular biology tests” section must be considered the use of a post-hoc test (e.g. Bonferroni or Holm-Bonferroni corrections) to give strength to the resulting data. Why no one was applied?We would like to apologize for this oversight that we made in statistical analysis. As asked also by an other reviewer we modified p-value using post-hoc test. When multiple comparison where performed between CTRL, DHA, DHA-D2 and DHA-D4 treatment, one‑way ANOVA analysis (or Kruskal-Wallis test for non-normally distributed data) followed by Bonferroni (or Dunn) post‑hoc test were performed to evaluate the significance between groups. P<0.05 was considered to indicate a statistically significant difference. Experimental section on statistical analysis was changed. The figures were changed all along the manuscript to adjust the new results obtained for significativity (#, ## or ###). The significance was lost between the comparison of four results: in Fig. 1C, Fig. 2, Fig. 3B and Fig. 5D. The manuscript was modified to be in agreement with those new comparisons. However those changes did not modify the main results and conclusions of the manuscript.
Even if the manuscript is well written, it requires minor English revisions.The manuscript was reviewed by native Irish researcher (Dr. Patrick Carroll) in addition to all co-authors, to eliminate English mistakes.
Reviewer 2 Report
The authors have examined the efficiency of selective deuterium incorporation at bis-allylic positions of DHA, to decrease toxicity and lipid peroxidation in ARPE-19 cells, under oxidative conditions. This study is interesting. However, the reviewer has some comments as follows;
Major concerns
An increase of cell viability by DHA treatment has been shown in figure 1 B2. Although the authors argue that the increase indicates a cytoprotective effect of DHA, it seems the basis is lacking. The authors should present the data on which DHA demonstrates cytoprotective effects in their experimental system. It seems that the effect of IP-DHAs on lipid peroxidation status is weaker than the effect of free DHAs. The authors should explain the gap of the effects between free and IP-DHAs. The authors should show the statistical results by multiple comparisons, although Student’s t-test or Mann-Whitney test have been used for all result in this manuscript.
Minor comments
The authors have described that “D4-DHA shows a significant reduction of lipid peroxidation (around 10%) compared to cells subjected to H2O2 without PUFA treatment” in line 404 - 405. If there is a significant difference between untreated cells and D4-DHA treated cells, the results of statistical processing should be shown.
Author Response
Reviewer 2:
First, we thank reviewer 2 for his/her time in reviewing this manuscript. The comments were much valuable and we have revised the manuscript accordingly and as much as possible. For more clarity, changes related to all reviewers comments were highlighted in green in the manuscript; changes related to English revision were highlighted using “Track Changes" function in Microsoft Word.
The authors have examined the efficiency of selective deuterium incorporation at bis-allylic positions of DHA, to decrease toxicity and lipid peroxidation in ARPE-19 cells, under oxidative conditions. This study is interesting. However, the reviewer has some comments as follows;
Major concerns
An increase of cell viability by DHA treatment has been shown in figure 1 B2. Although the authors argue that the increase indicates a cytoprotective effect of DHA, it seems the basis is lacking. The authors should present the data on which DHA demonstrates cytoprotective effects in their experimental system.We agree with reviewer 2, that additional experiments must been done to speak about « cytoprotection » in this case. In those experiments we only verify that the increase of DO and cell viability was not due to an increase of cell proliferation. Thus we changed the sentence in the result and discussion sections, replacing “cytoprotection” by “improvement of cell viability”.
It seems that the effect of IP-DHAs on lipid peroxidation status is weaker than the effect of free DHAs. The authors should explain the gap of the effects between free and IP-DHAs.We agree with reviewer 2. One of the explanation of the difference of protection observed between free deuterated PUFA and deuterated IP-DHA lipophenol could be a different rate of membrane incorporation, which could be lower using IP-DHA. However we did not performed experiment to confirm this point. A sentence was added in the discussion to present this hypothesis.
The authors should show the statistical results by multiple comparisons, although Student’s t-test or Mann-Whitney test have been used for all result in this manuscript.We would like to apologize for this oversight that we made in statistical analysis. As asked also by an other reviewer we modify p-value using post-hoc test. When multiple comparison where performed between CTRL, DHA, DHA-D2 and DHA-D4 treatment, one‑way ANOVA analysis (or Kruskal-Wallis test for non-normally distributed data) followed by Bonferroni (or Dunn) post‑hoc test were performed to evaluate the significance between groups. P<0.05 was considered to indicate a statistically significant difference. Experimental section on statistical analysis was changed. The figures were changed all along the manuscript to adjust the new results obtained for significativity (#, ## or ###). The significance was lost between the comparison of four results: in Fig. 1C, Fig. 2, Fig. 3B and Fig. 5D. The manuscript was modified to be in agreement with those new comparisons. However those changes did not modify the main results and conclusions of the manuscript.
Minor comments
The authors have described that “D4-DHA shows a significant reduction of lipid peroxidation (around 10%) compared to cells subjected to H2O2 without PUFA treatment” in line 404 - 405. If there is a significant difference between untreated cells and D4-DHA treated cells, the results of statistical processing should be shown.
Using student t-test the reduction of lipid peroxidation using D4-DHA vs No DHA under H202 (Fig. 2), was significant (13%) with statistical analysis giving ##p < 0.01 versus H2O2 treated cells. However using the multiple comparison statistical analysis, this reduction was not any more significant, thus the sentence in the result section and in the discussion section were removed.
Reviewer 3 Report
In the cell culture experiments, why the control group are under serum starvation (1%FBS) and not in 10% FBS?
“I really like the manuscript but I have some questions and comments:
1. I would recommend mentioning the passage number of the cells.
2. In the cell culture experiments, why the control group are under serum starvation (1%FBS) and not in 10% FBS?
3. Why H2O2 is used as an oxidative insult? How you set 600µM as an experimental condition? Did you try with lower doses? Did you try higher concentration in a shortened period of time (acute doses)?
4. Did you see any morphological change in the ARPE-19 cells?
5. Did you try to use the ARPE-19 cells in the differentiated state using inserts in the plate of culture?”
Author Response
Reviewer 3:
First, we thank reviewer 3 for his/her time in reviewing this manuscript. The comments were much valuable and we have revised the manuscript accordingly and as much as possible. For more clarity, changes related to all reviewers comments were highlighted in green in the manuscript; changes related to English revision were highlighted using “Track Changes" function in Microsoft Word.
“I really like the manuscript but I have some questions and comments:
I would recommend mentioning the passage number of the cells.ARPE-19 cells were cultured and used up a maximum of 10 passages, a sentence was added in the experimental section.
In the cell culture experiments, why the control group are under serum starvation (1%FBS) and not in 10% FBS?In the cell experiments, the data are expressed as the percentage of untreated cells (not treated by DHAs or IP-DHA), thus the control cells represent the cells under oxidative stress (H2O2 or serum starvation 1%) without DHAs or lipophenol treatment. The use of 10% FBS for control would not have represented the correct control since 1% FBS is actually used as stress condition, and as show in the figure S1 and S2 (SI), the percentage of serum influences deeply the oxidative status of the cells.
In addition, in this work we did not choose to work with 10% FBS because DHA was not toxic at all using this FBS concentration. Moreover, using 10% FBS, production of ROS was lowered in the presence of H202 and really high concentration of H2O2 had to be used to observe decrease in cell viability.
Why H2O2 is used as an oxidative insult? How you set 600 µM as an experimental condition? Did you try with lower doses? Did you try higher concentration in a shortened period of time (acute doses)?H2O2 was used to induce stronger oxidative stress compared to serum starvation and to produce higher amount of ROS. tBuOOH would have been used also, anyway we followed previous publication using H2O2 to generate oxidative stress in ARPE-19 (Johansson et al. 2015 for exemple). The concentration of H2O2 was chosen regarding result of experiment showed in figure S3, (Fig. S3. Toxicity of H2O2 in 1% FBSM on ARPE-19 cell). Lower doses of H202, 300 to 400µM reduced only marginally the cell viability. Using 500 µM the viability was reduced to 60-70%. The concentration 600 µM was selected as stronger oxidative stress condition because it caused strong reduction of cell viability (30-40%) without killing all the cells, and because ROS production did not increased using higher H2O2 concentrations. We did not experiment variation in H2O2 treatment duration.
Did you see any morphological change in the ARPE-19 cells ?The Fig. 1A shows the morphological changes observed before and after the used of the toxic concentration of DHA. After the use of toxic concentration of PUFA, the cells present more elongated structure and present also less adherence between themselves. A sentence was added in the text to highlight this point.
Using H202 treatment (Fig. not show), we observe kind of stress fiber.
Did you try to use the ARPE-19 cells in the differentiated state using inserts in the plate of culture?”No, this is something interesting but we did not performed such experiments, because ARPE-19 cells responded properly to mild and strong oxidative conditions and to natural and deuterated DHA.
Round 2
Reviewer 1 Report
After suggested revisions, applied by authors, the manuscript results suitable for publication.